# Dietary Intake of Adult Residents in Luxembourg Taking Part in Two Cross-Sectional Studies—ORISCAV-LUX (2007–2008) and ORISCAV-LUX 2 (2016–2017)

**DOI:** 10.3390/nu13124382

**Published:** 2021-12-07

**Authors:** Farhad Vahid, Alex Brito, Gwenaëlle Le Coroller, Michel Vaillant, Hanen Samouda, Torsten Bohn

**Affiliations:** 1Nutrition and Health Research Group, Population Health Department, Luxembourg Institute of Health, 1445 Strassen, Luxembourg; Farhad.vahid@lih.lu (F.V.); hanene.samouda@lih.lu (H.S.); 2Laboratory of Pharmacokinetics and Metabolomics Analysis, Institute of Translational Medicine and Biotechnology, I.M. Sechenov First Moscow State Medical University, 119991 Moscow, Russia; abrito@labworks.ru; 3World-Class Research Center “Digital Biodesign and Personalized Healthcare”, I.M. Sechenov First Moscow State Medical University, 119991 Moscow, Russia; 4Competence Center in Methodology and Statistics, Luxembourg Institute of Health, 1445 Strassen, Luxembourg; Gwenaelle.LeCoroller@lih.lu (G.L.C.); michel.vaillant@lih.lu (M.V.)

**Keywords:** dietary habits, food groups, calorie intake, vitamins, minerals, beta-carotene, sugar-sweetened beverages, exploratory factor analysis

## Abstract

Background: A balanced diet is an important lifestyle component and has been associated with a reduced risk of chronic diseases. Objectives: To assess dietary intake of adult residents in Luxembourg taking part in two population-based cross-sectional studies (ORISCAV-LUX, 2007–2008 and ORISCAV-LUX 2, 2016–2017). Methods: Dietary intake of the study participants (1242 in 2007/08 and 1326 in 2016/17), 25–69 years old, were evaluated using food-frequency questionnaires (134 items in 2007/2008 and 174 items in 2016/2017) according to the French ANSES-CIQUAL food composition database. Both food-group- and nutrient-based analyses were conducted. Results: Dietary patterns in ORISCAV-LUX 2, 2016–2017, were characterized by an increase in the estimated marginal means (EMM) of the intake of energy, total fat, saturated fatty acids, alcohol, and decreased EMM of total carbohydrates, magnesium, and calcium compared to 2007/08. We also observed an increased EMM of the intake of protein-rich food items and ready-to-eat foods/fast foods, together with a decreased intake of grains, dairy products, and vegetables (all *p*-values <0.05, linear mixed models). The intake of most micronutrients was stable or slightly increased in ORISCAV-LUX 2 vs. ORISCAV-LUX, except for the drop in magnesium and calcium, and generally met recommendations, in particular, EFSA population reference intakes (PRI), except for vitamin D. Conclusions: Though most micronutrient recommendations were met, nutrient consumption in terms of high energy, total fat, and sodium, as well as low carbohydrates, were not aligned with recommendations for balanced eating.

## 1. Introduction

Dietary patterns are an important nutritional and lifestyle component [1]. Poor dietary habits, such as over- or under-consumption of calories and macronutrients and a low intake of certain micronutrients or nonessential constituents, such as dietary fiber and secondary plant compounds, including carotenoids and polyphenols, have been related to several chronic noncommunicable diseases, such as cardiometabolic, neurodegenerative, and autoimmune diseases, as well as mental illnesses [1] and cancer [2]. Dietary patterns and food choices are influenced by various factors, such as age, gender, socioeconomic factors, and neurophysiological variations [3]. However, far from being fixed, these patterns might change over time at the population level due to changes in the population composition, sociodemographic factors, altered nutritional knowledge, and possibly food unavailability or inaccessibility, among others [4]. As improvements in dietary habits have been shown to be an important factor influencing overall health and all-cause mortality [5,6,7], monitoring dietary patterns is an important aspect for studying changes related to population health.

Luxembourg is characterized by a large percentage (about 50% of a total of ca. 650,000) of residents with foreign origin, possibly contributing to the diversity of eating habits [8]. Approximately 14.3% [8] of the population is 65 years old or older, a slightly higher fraction than in other European countries (12%) [9]. The country is also characterized, like many other Westernized countries, by a high prevalence of noncommunicable chronic diseases and related health conditions, potentially, although not exclusively, associated with a poor nutritional status, including prediabetes (25.6%) and diabetes (6.5%), overweight (37.3%) and obesity (20.6%), depression (21.6%), hypertension (31%), dyslipidemia (30.4%), cancer (3.6%), dementia (3.8%), cognitive impairment (26.1%), and Parkinson’s disease (0.2%) [10,11,12,13].

In Luxembourg, the dietary patterns, similar as to other Western-type diets [14], have been relatively high in saturated fat (18.4%), ready-made meals (7%), and alcohol consumption (13 L annual per capita) [15], all of which have been associated with cardiometabolic disturbances, such as a high incidence of metabolic syndrome, although this has been investigated in a cross-sectional study not adjusted for additional exposure [16]. In contrast, the Mediterranean dietary pattern, rich in unsaturated fats, whole grains, green leafy vegetables, fruits, and legumes, has been less frequently associated with various types of cancer and cardiometabolic diseases [17,18,19]. In general, dietary guidelines recommend choosing various vegetables, fruits, pulses, whole grains, and consuming a minimum amount of free sugars, processed/smoked meat, salt, and trans- and saturated fats, especially prevalent in ready-to-eat meals [20,21]. However, in Luxembourg, according to an earlier study in 2006/07, about 65% of the individuals did not reach the recommendations for dietary fiber intake [22], and about 50% did not consume five portions of 80–100 g of fruits and vegetables per day [22]. In addition, Luxembourg residents were ranked the highest meat consumers worldwide (136 kg per capita) in 2007 [23], with a large number of individuals regularly consuming ready-to-eat meals [16]. General dietary trends in the past years in most Westernized countries included tendencies for lower carbohydrate consumption [24], higher meat and processed food intake [25], but also leaning toward more organic/bio-foods [26].

Food choices, as a critical component in the overall dietary patterns, are complex and influenced by various factors [27]. One of the crucial goals of population-based longitudinal or repetitive cross-sectional studies is to monitor dietary changes to investigate food intake trends [28]. Such investigations are an important base for developing improved public health policy approaches [29]. This study was designed to investigate the changes in dietary patterns and habits concerning the intake of food groups, macro- and micronutrients, as well as non-nutrient compounds, in adult residents in Luxembourg taking part in two population-based, cross-sectional studies over the past decade (ORISCAV-LUX, 2007–2008 [8] and ORISCAV-LUX 2, 2016–2017 [30]).

## 2. Participants and Methods

### 2.1. Study Population and Design

The complete description of the study population and methods has previously been published in 2010 and 2019 [30]. Briefly, the Observation of Cardiovascular Risk Factors in Luxembourg (ORISCAV-LUX) surveys included two cross-sectional studies in adults residing in Luxembourg. In the original ORISCAV-LUX survey (2007–2008) [8], N = 1432 participants were included by a systematic random sampling procedure. In the original ORISCAV-LUX 2 survey (2016–2017) [30], N = 1558 participants were included by an initial baseline sampling plus complimentary sampling. A total of 660 individuals participated in both studies. The age ranges of the study participants in the original surveys were 18–69 years for ORISCAV-LUX and 25–79 years for ORISCAV-LUX 2.

In the present analysis, the same age ranges, 25–69 years, were retained to enable an accurate comparison between the two surveys (ORISCAV-LUX: N = 1242; ORISCAV-LUX 2: N = 1326). The participants were randomly selected based on sociodemographic attributes, including the district of residence, age, and gender. After a telephone appointment, the study participants were invited to take part in the surveys in the nearest study center from their domicile. During the study appointment, the study investigator gave the participants all the information related to the study, including the aim of the research project and the study protocol. The study participants received comprehensive guidance on the survey, including the general information questionnaire and the food frequency questionnaires (FFQ). In addition to the FFQ, selected parameters on anthropometric, demographic, and socioeconomic factors were collected. All the participants were duly informed and consented to take part in the study. The study design and information collected were approved by the National Research Ethics Committee (CNER) and the National Commission for Private Data Protection (CNPD).

### 2.2. Assessment of Dietary Intake

The dietary intake data were extracted from a validated quantitative FFQ [31]. In the ORISCAV-LUX study, a 134-item FFQ was used [31]. The FFQ was divided into nine food groups: 14 carbohydrate-related questions, 13 related to fruits, 13 to vegetables, 18 to meat–poultry–fish items, 11 to ready-made meals (prepared dishes), 22 to dairy products, 16 to fats (for spreading, cooking, and seasoning), 14 to drinks and beverages, and 13 to miscellaneous items. Miscellaneous items included jam, chocolate, peanut butter, dry biscuits, ice cream products, jellified desserts, sugar, and cocoa. The study participants indicated the portion size and frequency of all consumed beverages and food items on a scale ranging from “never or rarely”, “two or more times/day”, “once a day”, “3 to 5 times/week”, “1 to 2 times/week”, and “1 to 3 times/month”. The macro- and micronutrient intake was calculated by multiplying each food item’s consumption frequency by the specific nutrient content of each portion. Portion size images were used to accurately identify the portion sizes of all the consumed food and beverage items.

Similarly, in ORISCAV-LUX 2, a validated quantitative 174-item FFQ was used [32]. In fact, more questions about certain food items were asked in order to increase the accuracy in the second wave. For example, in the first wave, the question was about the total amount of “butter” consumed, while, in the second wave, the question was divided into two parts: “unsalted butter” and “lightly salted or salted butter”. The FFQ in the second wave comprised nine food groups, including 16 carbohydrate-related items, 12 fruit items, 13 vegetable items, 26 meat–poultry–fish items, 17 ready-made meal items, 22 dairy product items, 28 fat items, 21 drink and beverage items, and 18 miscellaneous items. The methods for completing the FFQ and extracting the data and the food database used [33] were similar for the two waves of the survey [33].

The amount of macro- and micronutrient intake was converted into a daily consumption and reported as median and interquartile range. For this purpose, the macro- and micronutrient intake amounts were obtained by linking the consumed food/beverage items with the ANSES-CIQUAL French Food Composition Table database [33]. The total energy was obtained as the sum of 37 kJ/g (9 kcal/g) for fat, 29 kJ/g (7 kcal/g) for alcohol, 17 kJ/g (4 kcal/g) for protein, 17 kJ/g (4 kcal/g) for carbohydrates (except for polyols), 13 kJ/g (3 kcal/g) for organic acids, 10 kJ/g (2.4 kcal/g) for polyols, and 8 kJ/g (2 kcal/g) for dietary fiber.

### 2.3. Anthropometric Measures

A trained research nurse performed the anthropometric measures of weight and height. Body mass index (BMI) was calculated. The height (cm) and body weight (kg) were measured in a slight dress without shoes. The participants’ BMI was estimated as weight in kg divided by the square of height in meters (kg/m^2^).

### 2.4. Demographic and Socioeconomic Factors

Age, gender, marital status, education, job, income, and the number of persons living in the same household were obtained from the General Information Questionnaire.

### 2.5. Data Management

From all the enrolled participants (ORISCAV-LUX = 1432, ORISCAV-LUX 2 = 1558), only the data of the participants who had completed the FFQ were considered in the present analysis. In this regard, 80 participants from ORISCAV-LUX and 127 participants from ORISCAV-LUX 2 were excluded due to the lack of FFQ data. From the 1352 participants who completed the FFQ in ORISCAV-LUX and the 1431 participants who completed the FFQ in ORISCAV-LUX 2, 110 participants under 25 years (ORISCAV-LUX) and 105 participants over 69 years (ORISCAV-LUX 2) were excluded to obtain the same age-range groups in the present paper.

Finally, the data of 1242 participants from ORISCAV-LUX and 1326 participants from ORISCAV-LUX 2 who completed the FFQ and were in the same age ranges were included in our analyses (see the flowchart of participants, Figure 1).

Missing data: For the variable “income”, we grouped the missing data into “did not answer”. Therefore, no other participants were excluded from the analysis due to missing data in sociodemographic variables (Table 1). The replacement of missing data by the means or their imputation was not considered as a suitable strategy as data were not to be missing at random.

### 2.6. Statistical Analyses

The normality of the data distribution and equality of variance were measured by Q–Q normality plots and the Kolmogorov–Smirnov test and box plots, respectively. A log transformation was performed for the non-normally distributed data.

Since about 45% of the participants in the second survey also participated in the first survey, linear mixed model (LMM) analyses were performed on log-transformed values to compare the estimated marginal means (EMMs) of the energy and macro- and micronutrient intake between the 2 surveys. LMMs included random intercepts for subjects and fixed effects for ORISCAV-LUX vs. ORISCAV-LUX 2, and adjustment for age at baseline, gender, marital status, education, job, income, and number of persons living in the same household. The LMMs, using an unstructured variance–covariance matrix, enabled the post hoc comparison of the estimated marginal means (EMMs) of the energy and macro- and micronutrient intake between the two surveys, and also according to the gender. A post hoc test (Tukey’s) was used. In order to decrease the false discovery rate due to the LMMS performed for each dietary parameter, we applied the Benjamini–Hochberg adjustment for multiple comparisons. The EMMs adjusted for age, gender, education, occupation (job), marital status, number of persons living in the household, and income were reported. In addition to the EMM and 95% confidence interval (95% CI), the raw data were reported as median and interquartile ranges. A *p*-value of 0.05 was considered as significant. The EMMs were also reported, adjusted only for age and gender, as a supplementary analysis. We also used the exploratory factor analysis (EFA) method to identify dietary patterns (2 major components), using the data from the FFQ, organized into 12 major food groups (Figure 2). Absolute values > 0.30 were considered to have a significant role in the components. Small coefficients below this value were suppressed.

A comparison of the average intake of macro- and micronutrients of the study participants in the two surveys with the recommended values published by the World Health Organization (WHO), European Food Safety Authority (EFSA, PRI), United States Department of Agriculture (USDA, RDA) dietary guidance, British Nutrition Foundation (BNF), and German-(D), Austrian-(A), and Swiss (CH) (DACH) reference values was also carried out. The SPSS statistical software (IBM SPSS statistics 25.0, IBM Corp., Armonk, NY, USA) was used for the statistical analyses.

## 3. Results

Overall, 51% of the participants in the ORISCAV-LUX and 53.4% of the participants in the ORISCAV-LUX 2 were women. The mean age was 46.3 ± 11.6 years in the ORISCAV-LUX and 49.5 ± 10.0 years in the ORISCAV-LUX 2. The general characteristics of the study participants are presented in Table 1.

The EMM obtained from the linear mixed models, as well as the median and interquartile ranges of total energy, water, alcohol, and macronutrient intake of participants, are presented in Table 2. There was a significant increase in the EMMs of total energy intake, total water, total protein, animal protein, total fat, cholesterol, polyunsaturated fatty acids (PUFA), saturated fatty acids (SFA), monounsaturated fatty acids (MUFA), linoleic acid, alpha-linolenic acid, and alcohol, and a significant decrease in the EMMs of total carbohydrate, simple sugar, and added sugar intake in ORISCAV-LUX 2 compared to ORISCAV-LUX.

Regarding micronutrient intake (Table 3), a significant reduction in the EMMs of magnesium and calcium intake was observed in ORISCAV-LUX 2 compared to ORISCAV-LUX.

The distribution of the participants’ food group intake is shown in Table 4. A significant decrease in the EMM of grains, vegetables, starchy vegetables, dairy products, and sugary products intake was found when comparing ORISCAV-LUX to ORISCAV-LUX 2, along with a significant increase in the EMM of protein-rich foods, ready-to-eat and fast food, lipids, noncaloric beverages, and alcoholic beverages. The same models for macro-, micro-nutrients and food groups adjusted only for age at baseline and gender are presented in Appendix A

The within- and between-group comparisons of macronutrient (Table 5) and micronutrient (Table 6) intakes based on gender groups showed that men consumed significantly more energy, fat, proteins, total carbohydrates, cholesterol, total fiber, and alcohol compared to women in both ORISCAV-LUX and ORISCAV-LUX 2. The intake of most micronutrients (except folate, vitamin E, vitamin C, and calcium) were also lower in women compared to men in both waves of ORISCAV. In addition, there was a significant increase in the intake of total energy, total fat, and total alcohol in men participating in ORISCAV-LUX2 compared to men in ORISCAV-LUX. Similar significant increases were seen in women in ORISCAV-LUX 2 compared to ORISCAV-LUX.

On the other hand, there was a significant decrease in consumed total carbohydrates in both men and women in ORISCAV-LUX 2, compared to ORISCAV-LUX (Table 5). In parallel with the total increased fat intake, there was a significantly higher intake of fat-soluble vitamins (A, D, and E) in both men and women participating in ORISCAV-LUX 2 compared to ORISCAV-LUX. In accordance with the reduction in grains and dairy product consumption in ORISCAV-LUX 2 compared to the ORISCAV-LUX, the intake of calcium and magnesium showed a significant decrease in both genders in ORISCAV-LUX 2 (Table 6).

Finally, Table 7 displays the comparison of the average intake of macro- and micronutrients of the study participants in the two surveys with the recommended values published by the World Health Organization (WHO), European Food Safety Authority (EFSA), United States Department of Agriculture (USDA) dietary guidance, British Nutrition Foundation (BNF), and German-(D), Austrian-(A), and Swiss (CH) (DACH) reference values.

Adults residing in Luxembourg and participating in ORISCAV-LUX and ORISCAV-LUX 2 showed an intake of vitamin D, fiber, and folate below several recommendations, and a higher than recommended sodium and total fat intake.

## 4. Discussion

The present work used data from two large cross-sectional studies, ORISCAV-LUX (2007/2008) and ORISCAV-LUX 2 (2016/2017). We described the dietary patterns and their changes during the last decade (2007–2017) in adult residents in Luxembourg having taken part in the surveys. Our results highlight dietary changes over approximately 10 years amongst the study participants, with significant differences in the amount of some consumed macronutrients and micronutrients and underlying food groups. Most notably, total fat intake, MUFA, SFA, PUFA (including eicosapentaenoic acid, docosahexaenoic acid, and docosapentaenoic acid), cholesterol, alcohol, and total energy intake in men and women did increase significantly over the past decade (Table 2). In contrast, total carbohydrate, magnesium, and calcium intakes were significantly reduced (Table 3).

Regarding major food groups, there was a decrease over time in the intake of grains, vegetables, and dairy products. In contrast, the intake of protein-rich foods, ready-to-eat meals, fats, noncaloric beverages, and alcoholic beverages increased during the studied period (Table 4).

The strongest intake increases during this approximately 10-year period were seen for alcohol, ready-made meals, total fat, and SFA (Table 2 and Table 3). Compared to dietary recommendations (Table 7), the intake of total fat, energy, SFA, and sodium appeared relatively high. Especially in conjunction with the increased intake of ready-to-eat foods/fast foods that also included processed foods, such as meat products, such trends have been associated in the literature with a high incidence of type 2 diabetes and other cardiometabolic diseases [34,35,36,37]. However, these associations are typically based on cross-sectional studies, and other lifestyle factors could confound such relations [38]. Despite the fact that Luxembourg, similar to other countries, is engaged in health promotion programs to stimulate healthy eating [39], it appears that health-promotion-oriented measures were insufficient to turn the tide of poor dietary patterns. In line with these findings, when using the exploratory factor analysis to determine the main dietary components, two dietary patterns were obtained, which were either characterized by a rather Mediterranean pattern, rich in whole grains, fruits, vegetables, and dairy products, or a rather westernized pattern rich in starchy vegetables, animal-based proteins, fast foods, and fats (Figure 2). The first pattern would be in line with diets that have been associated with generally favorable health outcomes [40].

The estimated intake of most micronutrients appeared comparable over the years, or even increased (Table 3), with the exception of calcium and magnesium intake undergoing a significant decline during the past decade. Magnesium is an essential macro-mineral and a reduction in dietary intake of this mineral over the past decades has been reported for other countries, such as the US [41]. The intake of this micronutrient has been related in meta-analyses to the decreased incidence of type 2 diabetes, cardiovascular diseases, and all-cause mortality [42]. As magnesium is consumed partly within the grain/carbohydrate group and within vegetables, it is possible that its decline was related to the reduced intake of these food groups observed in the present study. Moreover, due to the lower consumption of food items from the dairy group, the decrease in calcium intake is conceivable and predictable. In addition to its importance in bone mass density [43], numerous studies have examined the association between low calcium intake and an increased risk of CVD. A population-based study for instance concluded that dietary calcium intake is associated with a decreased CVD risk [44].

Despite a significant reduction in the intake of grains and vegetables, dietary fiber intake did not significantly change. In a Europe-wide cohort study [45], including over half a million participants, researchers reported that fiber intake was associated with various types of cancers, with reduced fiber intake from fruit and vegetable sources as a major possible cause [45]. In addition, fiber and associated phytochemicals originating from a diverse intake of plant-based food items might positively affect gut microbiota diversity [46], which has also been inversely associated with several chronic diseases, including diabetes [47]. Despite dietary fiber intake being marginal compared to some intake recommendations, it was very close to reaching the 25 g/day as stipulated by EFSA (Table 7).

Despite these findings, the intake of a number of micronutrients has been increasing over the past decade. In parallel with a generally higher fat intake, the intake of multi-unsaturated fatty acids, including omega-3 fatty acids (EPA, DHA, and linolenic acid), also increased. These fatty acids have generally been related to anti-inflammatory processes [48] and have been correlated, e.g., to a lower incidence of coronary heart diseases [48]. Another positive aspect associated with the intake of higher amounts of dietary lipids is the increased intake of fat-soluble vitamins A, E, and D (Table 3). This increase has resulted in almost reaching the respective intake recommendations set by several health and nutrition-related organizations (Table 7). Moreover, despite the decreased vegetable intake, beta-carotene consumption slightly increased from 2007 to 2017. This contradictory result could be due to the intake of other beta-carotene sources, such as carrots, cabbages, and avocado, which, in our study, showed an increase in their intake (results not shown). Furthermore, possibly due to increased total energy and protein intake from animal sources, some water-soluble vitamins, such as niacin and pyridoxine, also increased significantly (Table 3), and their intake was generally in line with dietary recommendations (Table 7).

However, our study results showed that the dietary patterns in ORISCAV-LUX 2 are moving to a more Westernized-type diet, characterized by a higher intake of fat and alcohol and a lower total carbohydrate intake. In line with our findings, Marques-Vidal et al. reported similar results in the French-speaking part of Switzerland [49]. These results indicate that minor changes in dietary intakes and choices over time can significantly affect overall dietary patterns.

Differences between genders were also observed. Our study results showed that men consumed significantly more total energy, protein, total fat, cholesterol, and alcohol than women. To some extent, the higher intake might be attributed to the higher energy needs of men compared to women [50]). It was also found that men in the ORISCAV-LUX 2 consumed significantly higher amounts of total energy, animal-based protein, total fat, fat-soluble vitamins, SFA, MUFA, PUFA, cholesterol, sodium, and alcohol than men in the ORISCAV-LUX survey, but lower amounts of calcium and magnesium. Similar results were observed when comparing women in the two waves, except for sodium, where no significant difference was observed. According to other studies, in line with our findings, dietary patterns in men and women have changed in the last decade or so in other westernized countries [51]. Bédard et al. showed that men especially consumed more high-fat, high-protein, and ready-to-eat foods than women [52]. Somewhat contrarily, Macdiarmid et al., in a study in the UK, reported that associations between sugar and fat intake and BMI were different between men and women. They concluded that the consumption of products rich in fats and sugars might partly explain the higher BMI in women than men [53].

Moreover, in line with our results and with worldwide trends, Sánchez-Villegas et al., in a cohort study, showed that Spanish adult residents tended to consume more ready-to-eat foods and meals and increased their intake of processed foods, and thus consumed more saturated fat and more sodium but fewer vegetables, low-fat dairy products, and fruits over the past years [54]. As observed for ORISCAV-LUX 2, Bamia et al. reported that western dietary patterns, mainly including fat, animal-based protein, and fast foods, are also rising in the elderly Europeans [55]. They emphasized that these dietary patterns could be an essential contributing factor for various diet-related diseases, such as diabetes [55] and other inflammatory diseases. For instance, Harding et al., in the EPIC Cancer-Norfolk study, found a significant association between dietary fat and cholesterol intake and diabetes [56], though, in a pooled meta-analysis of cohort studies, only saturated fat intake was related to some types of cancer, not total fat and cholesterol intake [57]. Our results also indicate that grain intake, a rich source of soluble and insoluble fiber, has rather decreased over the past decade. According to the “Dietary Patterns Amongst Older Europeans” survey, fiber, one of the important characteristics of the Mediterranean dietary pattern, has been shown to be associated with a reduction in metabolic diseases, such as diabetes and hypertension [55], possibly due to their positive influence on the gut flora and increased formation of anti-inflammatory short-chain fatty acids [58]. A review article by Matthias et al. following a Mediterranean dietary pattern with recommended amounts of whole grains, fruits, and vegetables highlighted that these patterns were associated with a significant reduction in a number of cardiometabolic diseases, such as diabetes and hypertension [59]. Therefore, a higher intake of dietary fiber, as opposed to the 25 g/day in the present study, is desired.

Furthermore, Michelle et al. reported that Mediterranean dietary patterns were associated with a lower likelihood of developing obesity in people that are overweight, suggesting that improving the nutritional status might be part of the solution in tackling obesity or overweightness [60]. Both metabolically unhealthy obesity and metabolically unhealthy normal weight have been on the rise in Luxembourg [61] and other European countries in the past decades and were associated with inflammatory and oxidative stress processes. Similarly, Buckland et al. of the EPIC cohort survey reported that the Mediterranean dietary pattern was associated with reduced breast cancer and coronary heart disease [62].

Several factors have been highlighted as contributing to the changes in dietary patterns over time. The most important ones in literature were economic/social status, education, age, and gender [63,64,65]. However, what remains to be more fully explored is why dietary patterns have shifted toward rather less healthy attributes in westernized countries, if not globally. In general, it is believed that factors such as the globalization of the economy and food production, widespread advertising of fast-food companies, and a lack of physical activity play important roles [66,67,68,69]. In addition, with globalization, staple foods have been shifting from local to industrial products, which entail, to a large extent, low-cost and highly processed foods and, consequently, result in a deterioration in healthy dietary patterns [70]. On the other hand, increasing working hours (together with less time eating at home), and easy access to cheap and ready-to-eat meals have been highlighted as individual factors in the westernization of dietary patterns over time [66,67,68,69].

One of our study’s strengths is that it is the first survey to examine an example of a European country with a diverse demographic composition [71]. Another advantage of our study was using a validated FFQ, which allowed us to have a comprehensive interpretation of the study participants’ dietary intake, together with a geographically appropriate food database. Our study has a relatively high sample size given the total population of Luxembourg, which allowed us to perform analyses for different age and gender groups. However, ORISCAV-LUX 2 is not fully representative of the general adult population residing in Luxembourg. For example, the number of Portuguese participants in the second survey was lower than the number of Portuguese participants in the EHES (14.5%) study, which is considered representative of the general population and was conducted almost at the same time as ORISCAV LUX 2 [72]. Contrarily, ORISCAV-LUX (as ORISCAV-LUX 2), when comparing respondents vs. nonrespondents at baseline, can be considered representative regarding the place of residence [30], though not for other variables, such as age or education level.

As for other observational studies, our study had some limitations. One of the shortcomings was related to the use of two FFQ, with the second one being slightly more detailed. Recall bias is considered inevitable, as the FFQ inquired about food intake in the past 3 months. However, it seems that employing trained personnel might significantly reduce this bias [73]. As for all population-based longitudinal and cohort studies, another concern in our surveys is the quality of collected data in the two study waves, such as sample measurement by an accredited laboratory. Due to the generally small number of missing data (except for income) and percentage of completed questions, this limitation does not seem detrimental to the results of this study, despite nutrition playing a crucial role in these groups. Different dietary assessment tools (e.g., multiple 24-h recall methods or food records) may be more recommendable for those groups than FFQ [73]. Therefore, it is proposed that future studies focus on these groups to obtain a more comprehensive overview and formulate more targeted nutritional interventions, starting early in life.

## 5. Conclusions

As for other Westernized countries, adults in Luxembourg taking part in ORISCAV-LUX 2 have been consuming relatively high amounts of processed foods, animal-based products, and thus proteins, and also fat and sodium. Concomitantly, a trend appears to consume slightly fewer vegetables, below the recommended intake. It is acknowledged that, in addition to physiological needs, an array of other factors, such as access to food, taste, the influence of peers, neurophysiological pathways to food intake, and socioeconomic factors, along with health promotion and public health actions are essential for improving dietary habits and patterns and deserve more investigation. Meanwhile, the State of Luxembourg has taken further steps to improve population health by fostering a healthier diet, including efforts such as introducing the Nutri-Score labeling [74], and has also provided community-based training based on age and gender. These measures remain to be awaiting their efficiency; further large-scale efforts and interventions to produce more substantial and lasting effects are desired. Additional monitoring of dietary patterns, including the very young, is paramount to monitor population-based efforts to steer lifestyle patterns toward healthy directions and reduce possible associated diseases.

## Figures and Tables

**Figure 1 nutrients-13-04382-f001:**
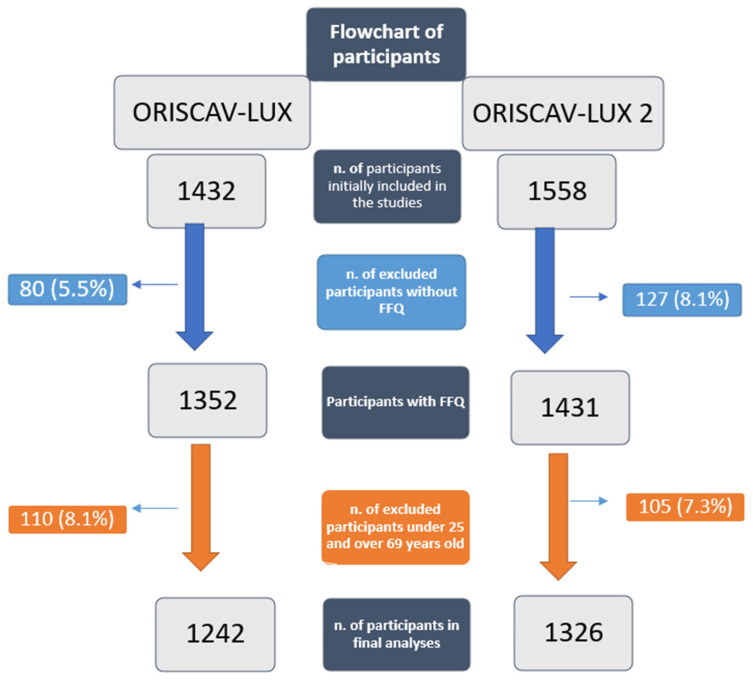
Flowchart of participants in ORISCAV-LUX and ORISCAV-LUX 2.

**Figure 2 nutrients-13-04382-f002:**
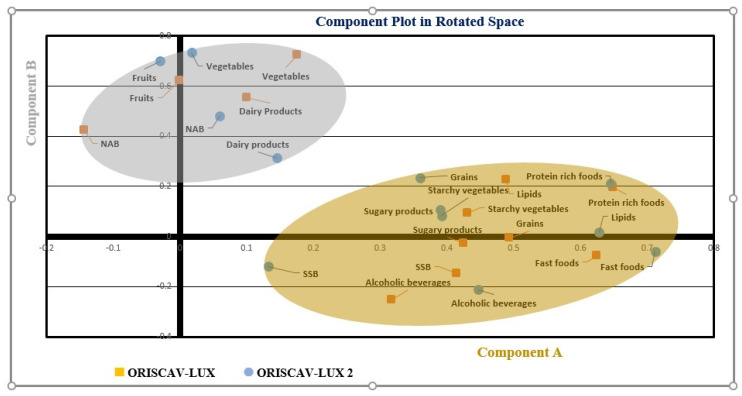
The exploratory factor analysis (EFA) was used to identify two major components, using the data from the FFQ organized into 12 major food groups. A component plot in rotated space for both ORISCAV-LUX and ORISCAV-LUX 2 is represented. Component A included fast foods (ready-to-eat meals), lipids, grains, starchy vegetables, alcoholic beverages, SSB (sugar-sweetened beverages), sugary products, and protein-rich foods. Component B included fruits, vegetables, NAB (nonalcoholic beverages), and dairy products.

**Table 1 nutrients-13-04382-t001:** Distribution of demographic, anthropometric, and socioeconomic characteristics of participants.

Variables	Mean ± SD (Minimum–Maximum) or Number (%)	
	ORISCAV-LUX (N = 1242)	ORISCAV-LUX 2 (N = 1326)
Age (year)	46.3 ± 11.6 (25.2–69.9)	49.5 ± 10.0 (25.2–69.9)
BMI (kg/m^2^)	26.9 ± 5.0 (16.1–51.2)	26.1 ± 4.7 (12.9–50.4)
Gender		
₋ Women	634 (51.1%)	709 (53.4%)
₋ Men	608 (48.9%)	617 (46.6%)
Marital status		
₋ Single ^a^	155 (12.5%)	150 (11.3%)
₋ Married	944 (76.0%)	984 (74.2%)
₋ Widow(er)	32 (2.6%)	162 (12.2%)
₋ Divorced or separated ^b^	111 (8.9%)	150 (11.3%)
Education		
₋ * No diploma	292 (23.5%)	169 (12.7%)
₋ Secondary education **	478 (38.5%)	463 (34.9%)
₋ Post-secondary education ***	364 (29.3%)	587 (44.3%)
₋ Did not answer	108 (8.7%)	107 (8.1%)
Occupation (Job)		
₋ Employed	835 (67.2%)	917 (69.1%)
₋ Unemployed ^c^	216 (17.4%)	141 (10.6%)
₋ Leave ^d^	170 (13.8%)	251 (19.0%)
₋ Did not answer	20 (1.6%)	17 (1.3%)
Income (EUR)		
₋ Less than 750	13 (1.0%)	4 (0.3%)
₋ 750 to 1499	49 (3.9%)	22 (1.7%)
₋ 1500 to 2249	143 (11.5%)	45 (3.4%)
₋ 2250 to 2999	195 (15.7%)	74 (5.6%)
₋ 3000 to 4999	381 (30.7%)	306 (23.1%)
₋ 5000 to 10,000	277 (22.3%)	466 (35.1%)
₋ More than 10,000	57 (4.6%)	111 (8.4%)
₋ Did not answer	127 (10.2%)	298 (22.5%)
Country of birth		
₋ Luxembourg	738 (59.4%)	778 (58.7%)
₋ Portugal	149 (12.0%)	110 (8.3%)
₋ Other European countries	246 (19.8%)	315 (23.3%)
₋ Non-European countries	109 (8.8%)	123 (9.3%)

Standard deviation = SD, body mass index = BMI. * Pre-primary and primary education. ** CATP—Certificate of Technical and Professional Aptitude, CITP—Certificate of Technical and Professional Initiation, CCM—Certificate of Manual Capability, Diploma for Completion of Secondary Technical Studies, Diploma for Completion of Secondary General Studies. *** Technician diploma, Bac +2 (BTS), Bac +3 (Bachelors/Degree), Bac +4 (Masters), Bac +5 and more (3rd Cycle, DEA, DESS, MBA, Masters, Ph.D., etc.), Diploma from a Grande Ecole, an Engineering School. ^a^ Single, never married, and never in a registered partnership. ^b^ Divorced, separated, separated but still legally married. ^c^ In school, university or in training, at home, unemployed or in search of employment. ^d^ Retired or in early retirement, on long-term leave.

**Table 2 nutrients-13-04382-t002:** Median (interquartile range) and estimated marginal means of participants’ total energy, alcohol, and macronutrient intake (*p*-values are based on linear mixed models, as further detailed in footnotes).

Parameter	Raw Values		Modeled Values		*p*-Value **
	Median (IQR)		Estimated Marginal Means (95% CI) *		
	ORISCAV-LUX	ORISCAV-LUX 2	ORISCAV-LUX	ORISCAV-LUX 2	
Total energy intake (kcal/day)	2213 (1142)	2374 (1136)	3.349 (3.340, 3.358)	3.379 (3.371, 3.387)	<0.001
Total water (g/day)	2920 (1353)	3083 (1250)	3.469 (3.461, 3.478)	3.482 (3.475, 3.490)	0.018
Total protein (g/day)	88.3 (46.4)	89.6 (45.3)	1.939 (1.929, 1.948)	1.954 (1.946, 1.963)	0.017
VSP (g/day)	26.6 (14.8)	26.7 (14.2)	1.427 (1.417, 1.437)	1.432 (1.422, 1.441)	0.474
Animal source protein (g/day)	56.8 (36.2)	60.6 (37.4)	1.746 (1.734, 1.757)	1.772 (1.760, 1.783)	<0.001
Total fat (g/day)	93.4 (57.1)	116.7 (64.7)	1.975 (1.964, 1.986)	2.066 (2.056, 2.075)	<0.001
SFA (g/day)	32.2 (21.4)	39.8 (23.1)	1.511 (1.499, 1.522)	1.594 (1.583, 1.604)	<0.001
MUFA (g/day)	39.2 (24.5)	46.8 (26.2)	1.598 (1.587, 1.610)	1.674 (1.664, 1.684)	<0.001
PUFA (g/day)	14.8 (10.0)	20.7 (13.4)	1.180 (1.168, 1.193)	1.323 (1.311, 1.335)	<0.001
Linoleic acid (g/day)	12.2 (8.5)	17.3 (11.7)	1.092 (1.079, 1.105)	1.240 (1.228, 1.252)	<0.001
Alpha-linoleic acid (g/day)	1.04 (0.74)	1.79 (1.40)	0.045 (0.031, 0.058)	0.265 (0.251, 0.279)	<0.001
Arachidonic acid (g/day)	0.15 (0.11)	0.19 (0.13)	−0.821 (−0.834, −0.808)	−0.720 (−0.733, −0.707)	<0.001
EPA (g/day)	0.12 (0.13)	0.20 (0.23)	−0.934 (−0.955, −0.912)	−0.769 (−0.794, −0.743)	<0.001
DPA (g/day)	0.06 (0.05)	0.08 (0.07)	−1.230 (−1.245, −1.214)	−1.133 (−1.151, −1.115)	<0.001
DHA (g/day)	0.18 (0.19)	0.28 (0.31)	−0.746 (−0.767, −0.726)	−0.591 (−0.614, −0.569)	<0.001
Cholesterol (mg/day)	310.5 (196.1)	356.5 (204.5)	2.485 (2.474, 2.497)	2.554 (2.543, 2.565)	<0.001
Total carbohydrates (g/day)	229.4 (125.8)	217.6 (110.1)	2.361 (2.351, 2.371)	2.341 (2.332, 2.350)	0.002
Simple sugars (g/day)	108.3 (71.5)	100.1 (59.6)	2.038 (2.025, 2.051)	1.995 (1.983, 2.006)	<0.001
Added sugars (g/day)	31.7 (32.9)	28.1 (27.6)	1.504 (1.485, 1.523)	1.433 (1.414, 1.451)	<0.001
Starch (g/day)	108.5 (68.0)	102.9 (61.7)	2.029 (2.018, 2.040)	2.019 (2.008, 2.030)	0.220
Total fiber (g/day)	22.9 (12.8)	23.1 (12.0)	1.367 (1.356, 1.377)	1.357 (1.347, 1.367)	0.178
Soluble fiber (g/day)	4.6 (2.6)	4.7 (2.4)	0.667 (0.666, 0.687)	0.662 (0.652, 0.673)	0.049
Alcohol (g/day)	4.1 (11.5)	5.6 (11.2)	0.641 (0.601, 0.682)	0.762 (0.729, 0.794)	<0.001

* Linear mixed model (based on log-transformed data) adjusted for age, gender, marital status, education, job, income, number of persons living in the same household. ** Benjamini–Hochberg correction was applied to all *p*-values: all *p*-values are displayed after this correction. Vegetable source protein = VSP, interquartile range = IQR, saturated fatty acids = SFA, monounsaturated fatty acids = MUFA, polyunsaturated fatty acids = PUFA, eicosapentaenoic acid = EPA, docosapentaenoic acid = DPA, docosahexaenoic acid = DHA.

**Table 3 nutrients-13-04382-t003:** Median (interquartile range) and estimated marginal means of micronutrient intake of participants (*p*-values are based on linear mixed models, as explained in footnotes).

Parameter	Raw Values		Modeled Values		*p*-Value **
	Median (IQR)		Estimated Marginal Means (95% CI) *		
	ORISCAV-LUX	ORISCAV-LUX 2	ORISCAV-LUX	ORISCAV-LUX 2	
Vitamin A (µg/day)	364.8 (399.7)	475.5 (340.3)	2.595 (2.578, 2.612)	2.670 (2.657, 2.684)	<0.001
Beta-carotene (µg/day)	4121 (3690)	4973 (4094)	3.637 (3.620, 3.654)	3.689 (3.673, 3.705)	<0.001
Vitamin D (µg/day)	2.6 (3.2)	5.1 (4.6)	0.410 (0.389, 0.430)	0.693 (0.677, 0.710)	<0.001
Vitamin E (mg/day)	13.8 (8.8)	18.4 (11.6)	1.151 (1.139, 1.163)	1.269 (1.258, 1.280)	<0.001
Vitamin C (mg/day)	135.1 (109.1)	145.1 (101.7)	2.129 (2.114, 2.144)	2.150 (2.136, 2.163)	0.049
Thiamine (mg/day)	1.55 (0.85)	1.53 (0.81)	0.184 (0.174, 0.194)	0.192 (0.183, 0.201)	0.221
Riboflavin (mg/day)	1.86 (1.04)	1.83 (0.96)	0.271 (0.261, 0.281)	0.269 (0.260, 0.278)	0.794
Niacin (mg/day)	21.2 (11.4)	23.0 (12.0)	1.325 (1.315, 1.334)	1.361 (1.352, 1.370)	<0.001
Pantothenic acid (mg/day)	5.29 (2.71)	5.85 (2.91)	0.720 (0.711, 0.730)	0.771 (0.762, 0.780)	<0.001
Pyridoxine (mg/day)	2.18 (1.18)	2.39 (1.22)	0.340 (0.330, 0.349)	0.381 (0.372, 0.390)	<0.001
Folate (µg/day)	351.0 (196.8)	349.6 (172.3)	2.546 (2.536, 2.557)	2.538 (2.529, 2.547)	0.220
Vitamin B12 (µg/day)	5.34 (4.35)	6.30 (4.54)	0.723 (0.709, 0.737)	0.798 (0.785, 0.812)	<0.001
Calcium (mg/day)	1047 (518.3)	933.6 (454.3)	3.022 (3.013, 3.032)	2.969 (2.961, 2.978)	<0.001
Iron (mg/day)	13.9 (7.1)	14.3 (6.7)	1.143 (1.133, 1.152)	1.154 (1.145, 1.162)	0.082
Iodide (µg/day)	143.8 (82.6)	154.9 (78.8)	2.157 (2.147, 2.167)	2.193 (2.183, 2.202)	<0.001
Magnesium (mg/day)	411.0 (177.2)	373.5 (161.0)	2.620 (2.612, 2.628)	2.574 (2.566, 2.581)	<0.001
Potassium (mg/day)	3575 (1638)	3526 (1547)	3.550 (3.541, 3.559)	3.543 (3.535, 3.551)	0.220
Phosphorus (mg/day)	1354 (686.6)	1330 (612.4)	3.134 (3.124, 3.143)	3.126 (3.118, 3.134)	0.220
Sodium (mg/day)	2332 (1878)	3333 (1957)	3.497 (3.487, 3.508)	3.531 (3.521, 3.541)	<0.001

* Linear mixed model (based on log-transformed data) adjusted for age, gender, marital status, education, job, income, number of persons living in the same household. ** Benjamini–Hochberg correction was applied to all *p*-values: all *p*-values are displayed after this correction.

**Table 4 nutrients-13-04382-t004:** Median (interquartile range) and estimated marginal means of food group intake of participants (*p*-values are based on linear mixed models, as explained in footnotes).

Parameter	Raw Values		Modeled Values		*p*-Value **
	Median (IQR)		Estimated Marginal Means (95% CI) *		
	ORISCAV-LUX	ORISCAV-LUX 2	ORISCAV-LUX	ORISCAV-LUX 2	
Grains (g/day)	196.7 (140.0)	119.1 (101.1)	2.275 (2.260, 2.289)	2.075 (2.085, 2.092)	<0.001
Fruits (g/day)	289.8 (315.5)	286.6 (268.2)	2.419 (2.395, 2.442)	2.414 (2.395, 2.434)	0.779
Vegetables (g/day)	261.6 (232.6)	216.4 (171.8)	2.427 (2.410, 2.444)	2.302 (2.286, 2.318)	<0.001
Starchy vegetables (g/day)	57.1 (82.8)	56.7 (60.7)	1.765 (1.743, 1.786)	1.725 (1.704, 1.745)	0.008
Protein-rich foods (g/day)	161.0 (118.1)	213.7 (147.9)	2.181 (2.166, 2.196)	2.322 (2.309, 2.335)	<0.001
Ready-to-eat and fast foods (g/day)	83.3 (87.9)	95.7 (103.8)	1.879 (1.858, 1.899)	1.948 (1.927, 1.969)	<0.001
Dairy products (g/day)	233.8 (254.0)	178.4 (199.7)	2.322 (2.299, 2.346)	2.163 (2.139, 2.188)	<0.001
Lipids (fats and oils) (g/day)	40.8 (37.4)	61.4 (51.5)	1.606 (1.588, 1.623)	1.768 (1.752, 1.784)	<0.001
Sugary products (g/day)	38.0 (46.7)	33.6 (41.4)	1.545 (1.519, 1.572)	1.495 (1.472, 1.518)	0.006
- NCB (g/day)	1515 (989.3)	1698 (1011)	3.131 (3.114, 3.148)	3.198 (3.186, 3.210)	<0.001
- SSB (g/day)	53.5 (237.2)	70.7 (233.3)	2.045 (2.005, 2.086)	2.074 (2.039, 2.109)	0.333
- Alcoholic beverages (g/day)	58.6 (172.3)	76.2 (157.4)	1.908 (1.874, 1.941)	1.955 (1.927, 1.983)	0.019

* Linear mixed model (based on log-transformed data) adjusted for age, gender, marital status, education, job, income, number of persons living in the same household. ** Benjamini–Hochberg correction was applied to all *p*-values: all *p*-values are displayed after this correction. Interquartile range = IQR, noncaloric beverages = NCB, sugar-sweetened beverages = SSB.

**Table 5 nutrients-13-04382-t005:** Within- and between-group comparisons * of macronutrients based on gender groups (*p*-values are based on linear mixed model) ^e^.

	ORISCAV-LUX				*p*-Value ^a^	ORISCAV-LUX 2				*p*-Value ^a^	Men W 1 vs. W 2	Women W 1 vs. W 2
	Men (n = 608)		Women (n = 634)			Men (n = 617)		Women (n = 709)				
	Median (IQR)	EMM (95% CI)	Median (IQR)	EMM (95% CI)		Median (IQR)	EMM (95% CI)	Median (IQR)	EMM (95% CI)		*p*-Value ^c^	*p*-Value ^d^
Total energy intake (kcal/day)	2435 (1242)	3.395 (3.375, 3.415)	2015 (1005)	3.313 (3.293, 3.333)	<0.001	2684 (1187)	3.432 (3.413, 3.452)	2133 (922)	3.336 (3.317, 3.354)	<0.001	<0.001	<0.001
Total water (g/day)	2944 (1370)	3.468 (3.451, 3.485)	2901 (1325)	3.451 (3.434, 3.469)	0.421	3162 (1389)	3.483 (3.465, 3.501)	3006 (1118)	3.466 (3.449, 3.483)	0.126	0.035	0.040
Total protein intake (g/day)	96.5 (50.3)	1.994 (1.974, 2.014)	80.1 (40.7)	1.894 (1.874, 1.914)	<0.001	102 (48.6)	2.006 (1.986, 2.026)	80.1 (39.1)	1.900 (1.880, 1.919)	<0.001	0.126	0.521
Vegetable protein (g/day)	29.1 (15.8)	1.489 (1.467, 1.510)	24.4 (13.3)	1.406 (1.384, 1.427)	<0.001	29.2 (15.5)	1.490 (1.468, 1.511)	24.7 (12.3)	1.410 (1.389, 1.431)	<0.001	0.906	0.656
Animal source protein (g/day)	64.1 (37.8)	1.787 (1.762, 1.812)	51.6 (32.0)	1.684 (1.659, 1.709)	<0.001	70.1 (40.4)	1.822 (1.795, 1.848)	53.4 (32.1)	1.694 (1.668, 1.720)	<0.001	0.001	0.890
Total fat (g/day)	99.8 (60.5)	2.000 (1.976, 2.024)	88.7 (52.5)	1.950 (1.926, 1.974)	<0.001	128 (66.7)	2.101 (2.078, 2.124)	108.9 (56.8)	2.028 (2.005, 2.050)	<0.001	<0.001	<0.001
SFA (g/day)	35.2 (24.3)	1.536 (1.511, 1.562)	30.0 (19.2)	1.476(1.451, 1.502)	<0.001	44.5 (25.1)	1.627 (1.601, 1.652)	36.2 (20.0)	1.545 (1.521, 1.569)	<0.001	<0.001	<0.001
MUFA (d/day)	41.7 (24.8)	1.619 (1.595, 1.644)	37.3 (24.0)	1.575 (1.550, 1.600)	<0.001	52.3 (27.7)	1.707 (1.683, 1.732)	43.2 (22.4)	1.633 (1.610, 1.657)	<0.001	<0.001	<0.001
PUFA (g/day)	15.4 (10.0)	1.196 (1.169, 1.223)	14.4 (9.6)	1.156 (1.128, 1.183)	<0.001	22.6 (13.9)	1.354 (1.326, 1.382)	19.6 (12.5)	1.287 (1.260, 1.314)	<0.001	<0.001	<0.001
- Linoleic acid (g/day)	12.5 (8.6)	1.105 (1.077, 1.133)	12.0 (8.3)	1.065 (1.036, 1.094)	<0.001	18.8 (12.0)	1.271 (1.242, 1.300)	16.2 (11.1)	1.200 (1.171, 1.228)	<0.001	<0.001	<0.001
- Alpha-linoleic acid (g/day)	1.08 (0.73)	0.065 (0.036, 0.095)	1.02 (0.77)	0.039 (0.009, 0.070)	0.295	1.89 (1.29)	0.303 (0.272, 0.334)	1.68 (1.45)	0.016 (0.228, 0.290)	0.003	<0.001	<0.001
- Arachidonic acid (g/day)	0.17 (0.13)	−0.746 (−0.775, −0.718)	0.13 (0.09)	−0.894 (−0.923, −0.864)	<0.001	0.22 (0.15)	−0.657 (−0.687, −0.628)	0.17 (0.11)	−0.792 (−0.821, −0.762)	<0.001	<0.001	<0.001
- EPA (g/day)	0.13 (0.13)	−0.868 (−0.917, −0.819	0.10 (0.13)	−0.988 (−1.038, −0.937)	<0.001	0.21 (0.24)	−0.705 (−0.758, −0.653)	0.18 (0.23)	−0.808 (−0.863, −0.753)	<0.001	<0.001	<0.001
- DPA (g/day)	0.07 (0.06)	−1.150 (−1.185, −1.115)	0.05 (0.04)	−1.298 (−1.334, −1.265)	<0.001	0.08 (0.08)	−1.064 (−1.101, −1.027)	0.07 (0.07)	−1.196 (−1.235, −1.157)	<0.001	<0.001	<0.001
- DHA (g/day)	0.20 (0.20)	−0.680 (−0.727, −0.634)	0.16 (0.18)	−0.792 (−0.840, −0.745)	<0.001	0.29 (0.33)	−0.537 (−0.585, −0.488)	0.26 (0.31)	−0.619 (−0.668, −0.570)	<0.001	<0.001	<0.001
Cholesterol intake (mg/day)	352 (220)	2.541 (2.516, 2.566)	279 (173)	2.433 (2.408, 2.459)	<0.001	395 (225)	2.600 (2.574, 2.625)	323 (169)	2.502 (2.477, 2.527)	<0.001	<0.001	<0.001
Total carbohydrates (g/day)	250 (134)	2.421 (2.399, 2.442)	210 (105)	2.337 (2.315, 2.359)	<0.001	240 (121)	2.406 (2.384, 2.428)	197 (92.2)	2.310 (2.289, 2.331)	<0.001	0.084	<0.001
Total fiber (g/day)	23.4 (13.1)	1.390 (1.367, 1.412)	22.6 (12.6)	1.364 (1.341, 1.387)	0.030	23.6 (12.4)	1.384 (1.361, 1.408)	22.8 (11.7)	1.360 (1.337, 1.383)	0.015	0.858	0.565
- Soluble fiber (g/day)	4.5 (2.5)	0.686 (0.662, 0.709)	4.7 (2.6)	0.679 (0.655, 0.703)	0.830	4.6 (2.4)	0.676 (0.652, 0.701)	4.7 (2.4)	0.673 (0.649, 0.969)	0.664	0.662	0.516
Alcohol (g/day)	8.2 (18.1)	0.728 (0.640, 0.816)	2.0 (5.7)	0.303 (0.209, 0.396)	<0.001	9.3 (15.6)	0.837 (0.750, 0.925)	3.3 (7.5)	0.466 (0.379, 0.552)	<0.001	<0.001	<0.001

^a^ Within-group comparison. ^c^ Between-group comparison, men. ^d^ Between-group comparison, women. ^e^ Linear mixed model adjusted for age, marital status, education, job, income, number of persons living in the same household. * Benjamini–Hochberg correction was applied to all *p*-values: all *p*-values are displayed after this correction. Saturated fatty acids = SFA, monounsaturated fatty acids = MUFA, polyunsaturated fatty acids = PUFA, eicosapentaenoic acid = EPA, docosapentaenoic acid = DPA, docosahexaenoic acid = DHA, interquartile range = IQR, marginal means = EMM based on log-transformed data, wave = W.

**Table 6 nutrients-13-04382-t006:** Within- and between-group comparisons* of micronutrients based on gender groups (*p*-values are based on linear mixed model) ^e^.

	ORISCAV-LUX					ORISCAV-LUX 2					Men W 1 vs. W 2	Women W 1 vs. W 2
	Men (n = 608)		Women (n = 634)		*p*-Value ^a^	Men (n= 617)		Women (n = 709)		*p*-Value ^a^		
	Median (IQR)	EMM (95% CI)	Median (IQR)	EMM (95% CI)		Median (IQR)	EMM (95% CI)	Median (IQR)	EMM (95% CI)		*p*-Value ^c^	*p*-Value ^d^
Vitamin A (µg/day)	412 (491)	2.662 (2.625, 2.698)	337 (306)	2.562 (2.526, 2.597)	<0.001	541 (375)	2.737 (2.703, 2.771)	425 (276)	2.640 (2.607, 2.674)	<0.001	<0.001	<0.001
Beta-carotene (µg/day)	3789 (3241)	3.609 (3.572, 3.645)	4427 (4195)	3.669 (3.632, 3.707)	<0.001	4693 (3733)	3.669 (3.630, 3.707)	5108 (4515)	3.727 (3.690, 3.763)	<0.001	<0.001	<0.001
Vitamin D (µg/day)	2.9 (3.4)	0.468 (0.425, 0.511)	2.3 (2.7)	0.362 (0.319, 0.406)	<0.001	5.6 (4.7)	0.733 (0.692, 0.774)	4.8 (4.4)	0.664 (0.623, 0.704)	<0.001	<0.001	<0.001
Vitamin E (mg/day)	13.7 (8.3)	1.137 (1.111, 1.163)	14.1 (9.0)	1.143 (1.117, 1.170)	0.208	20.3 (13.5)	1.295 (1.269, 1.321)	16.7 (9.5)	1.225 (1.200, 1.250)	<0.001	<0.001	<0.001
Vitamin C (mg/day)	128 (102)	2.130 (2.098, 2.162)	141 (120)	2.152 (2.119, 2.186)	0.038	141 (99.1)	2.161 (2.129, 2.194)	150 (103)	2.183 (2.152, 2.215)	0.093	0.009	0.086
Thiamine (mg/day)	1.7 (0.9)	0.234 (0.212, 0.255)	1.4 (0.7)	0.151 (0.129, 0.173)	<0.001	1.7 (0.9)	0.245 (0.223, 0.267)	1.4 (0.6)	0.154 (0.133, 0.175)	<0.001	0.110	0.469
Riboflavin (mg/day)	1.9 (1.1)	0.309 (0.287, 0.330)	1.7 (0.9)	0.250 (0.228, 0.272)	<0.001	2.0 (1.0)	0.315 (0.293, 0.337)	1.6 (0.8)	0.238 (0.217, 0.259)	<0.001	0.279	0.038
Niacin (mg/day)	23.8 (12.3)	1.373 (1.352, 1.393)	19.4 (9.6)	1.273 (1.253, 1.294)	<0.001	26.3 (13.2)	1.414 (1.393, 1.435)	20.2 (9.8)	1.302 (1.282, 1.322)	<0.001	<0.001	<0.001
Pantothenic acid (mg/day)	5.6 (2.8)	0.765 (0.744, 0.785)	5.0 (2.4)	0.699 (0.678, 0.720)	<0.001	6.4 (3.3)	0.819 (0.798, 0.840)	5.4 (2.4)	0.747 (0.727, 0.767)	<0.001	<0.001	<0.001
Pyridoxine (mg/day)	2.3 (1.2)	0.386 (0.365, 0.407)	2.0 (1.0)	0.305 (0.284, 0.327)	<0.001	2.6 (1.3)	0.434 (0.412, 0.455)	2.2 (1.0)	0.342 (0.321, 0.363)	<0.001	<0.001	<0.001
Folate (µg/day)	354 (203)	2.563 (2.541, 2.585)	347 (194)	2.547 (2.525, 2.570)	0.418	358 (179)	2.567 (2.545, 2.589)	340 (162)	2.541 (2.520, 2.563)	0.004	0.311	0.284
Vitamin B12 (µg/day)	6.0 (4.8)	0.795 (0.764, 0.826)	4.8 (3.8)	0.678 (0.647, 0.709)	<0.001	7.4 (4.9)	0.875 (0.844, 0.906)	5.5 (4.0)	0.746 (0.715, 0.777)	<0.001	<0.001	<0.001
Calcium (mg/day)	1043 (520)	3.029 (3.008, 3.050)	1048 (507)	3.024 (3.004, 3.045)	0.605	990 (473)	2.983 (2.963, 3.004)	904 (433)	2.960 (2.940, 2.981)	<0.001	<0.001	<0.001
Iron (mg/day)	15.4 (7.7)	1.192 (1.172, 1.212)	12.8 (6.5)	1.108 (1.088, 1.129)	<0.001	15.7 (7.5)	1.203 (1.182, 1.223)	13.3 (5.9)	1.120 (1.101, 1.140)	<0.001	0.104	0.174
Iodide (µg/day)	151 (80.9)	2.194 (2.171, 2.216)	134 (83.5)	2.143 (2.121, 2.166)	<0.001	166 (84.3)	2.235 (2.213, 2.257)	145 (71.7)	2.174 (2.152, 2.195)	<0.001	<0.001	<0.001
Magnesium (mg/day)	430 (192)	2.649 (2.631, 2.666)	391 (164)	2.604 (2.586, 2.621)	<0.001	404 (171)	2.611 (2.593, 2.629)	353 (146)	2.554 (2.536, 2.571)	<0.001	<0.001	<0.001
Potassium (mg/day)	3683 (1701)	3.583 (3.564, 3.602)	3378 (1531)	3.542 (3.522, 3.562)	<0.001	3721 (1635)	3.582 (3.562, 3.601)	3370 (1447)	3.540 (3.521, 3.559)	<0.001	0.928	0.354
Phosphorus (mg/day)	1483 (763)	3.178 (3.158, 3.198)	1280 (637)	3.104 (3.083, 3.124)	<0.001	1504 (694)	3.174 (3.154, 3.194)	1215 (544)	3.090 (3.071, 3.110)	<0.001	0.699	0.016
Sodium (mg/day)	3703 (2143)	3.549 (3.526, 3.572)	2895 (1524)	3.444 (3.421, 3.467)	<0.001	3894 (2143)	3.586 (3.562, 3.609)	2952 (1529)	3.462 (3.439, 3.484)	<0.001	<0.001	0.310

^a^ Within-group comparison. ^c^ Between-group comparison, men. ^d^ Between-group comparison, women. ^e^ Linear mixed model adjusted for age, marital status, education, job, income, number of persons living in the same household. * Benjamini–Hochberg correction was applied to all *p*-values: all *p*-values are displayed after this correction. Interquartile range = IQR, marginal means = EMM based on log-transformed data, wave = W.

**Table 7 nutrients-13-04382-t007:** Mean intake of macro- and micronutrients of participants in ORISCAV-LUX and ORISCAV-LUX 2 compared to recommended values.

	Wave 1		Wave 2		WHO ^a^		USDA ^ꝭ^		BNF ^a^		DACH		EFSA ^Δ^	
	M	W	M	W	M	W	M	W	M	W	M	W	M	W
Total energy intake (kcal/day) ^d^	2660	2191	2785	2264	2500	2000	ND	ND	2500 ^e^	2000 ^e^	2600	2000 ^e^	2000 ^e^	1800 ^e^
Total water (g/day)	3057	2995	3242	3103	3700 ^c^	2700 ^c^	3700	2700	ND	ND	2700	2700	2500 ^c^	2000 ^c^
Total protein (g/day) (%) ^t^	104 (15%)	84.4 (15%)	106 (23%)	84.1 (14%)	0.66 ^b^	0.66 ^b^	56	46	0.75 ^b^	0.75 ^b^	56	50	0.83 ^b^	0.83 ^b^
Total fat (g/day) (%) ^t^	109 (37%)	97.0 (40%)	135 (43%)	115 (46%)	20–35%	20–35%	ND	ND	35%	35%	30%	30%	30%	30
SFA (g/day) (%) ^t^	39.0 (13%)	33.4 (14%)	46.8 (15%)	38.8 (15%)	10%	10%	ND	ND	11%	11%	ND	ND	ND	ND
MUFA (g/day) (%) ^t^	45.8 (15%)	41.0 (17%)	55.2 (18%)	47.0 (18%)	15–20%	15–20%	ND	ND	ND	ND	ND	ND	ND	ND
PUFA (g/day) (%)t	17.4 (11%)	16.2 (7%)	25.0 (8%)	22.4 (9%)	6–11%	6–11%	ND	ND	ND	ND	ND	ND	ND	ND
Carbohydrate (g/day) (%) ^t^	282 (42%)	227 (41%)	257 (37%)	210 (37%)	55–75%	55–75%	13D	130	50%	50%	50%	50%	50%	50%
Total fiber (g/day)	25.3	24.2	25.4	24.3	38	25	38	25	30	30	30	30	25 ^c^	25 ^c^
Alcohol (g/day)	12.5	4.8	14.8	6.3	ND	ND	ND	ND	ND	ND	20	10	ND	ND
Vitamin A (µg/day)	549	437	603	490	600	500	900	700	700	600	1.0 ^g^	0.8 ^g^	750	650
Vitamin D (µg/day)	3.6	3.0	6.6	5.7	5 *	5 *	5 *	5 *	10	10	20 ^s^	20 ^s^	15 ^c^	15 ^c^
Vitamin E (mg/day)	15.4	15.6	21.7	18.8	10	7.5	15	15	ND	ND	14	12	13 ^c^	11 ^c^
Vitamin C (mg/day)	151	162	161	170	45	45	90	75	40	40	110	95	110	95
Thiamine (mg/day)	1.8	1.5	1.8	1.5	1.2	1.1	1.2	1.1	1.0	0.8	1.2	1.0	1.0	1.0
Riboflavin (mg/day)	2.1	1.9	2.1	1.8	1.3	1.1	1.3	1.1	1.3	1.1	1.3	1.1	1.6	1.6
Niacin (mg/day)	24.8	20.1	27.3	21.4	16	14	16	14	16	14	15	11	16	16
Pantothenic acid (mg/day)	6.0	5.2	6.7	5.7	5	5	5	5	ND	ND	6	6	5	5
Pyridoxine (mg/day)	2.5	2.1	2.8	2.3	1.7	1.5	1.3 *	1.3 *	1.4	1.2	1.6	1.4	1.7	1.6
Folate (µg/day)	374	372	383	363	400	400	400	400	200	200	300	300	330	330
Vitamin B12 (µg/day)	6.8	5.3	8.1	6.3	2.4	2.4	2.4	2.4	1.5	1.5	4.0	4.0	4.0 ^c^	4.0 ^c^
Calcium (mg/day)	1136	1111	1023	969	1000 *	1000 *	1000 *	1000 *	700	700	1000	1000	1000 ^c^	1000 ^c^
Iron (mg/day)	16.1	13.4	16.4	13.8	8	18 *	8	18 *	8.7	14.8 *	10	15/10 ^p^	11	16/11 ^p^
Iodide (µg/day)	162	147	177	155	200	150	150	150	140	140	190	150	150 ^c^	150 ^c^
Magnesium (mg/day)	453	410	419	371	260	220	420 ^l^	320 ^l^	300	270	350	300	350	300
Potassium (mg/day)	3886	3584	3860	3543	3400 ^c^	2600 ^c^	4700	4700	3500	3500	4000	4000	3500	3500
Phosphorus (mg/day)	1571	1350	1542	1284	700	700	700	700	550	550	700	700	550	550
Sodium (mg/day)	3937	3092	4152	3154	2000	2000	2300	2300	1600	1600	1500	1500	2000	2000

ORISCAV = Observation of Cardiovascular Risk Factors; WHO = World Health Organization; USDA = United States Department of Agriculture; BNF = British Nutrition Foundation; DACH = Germany (D), Austria (A), and Switzerland (CH) Reference Values; EFSA = European Food Safety Authority; ^ꝭ^ Dietary Reference Intakes (DRIs): recommended dietary allowances (RDAs), if unavailable, acceptable intake (AI) was used; ^a^ recommended nutrient intake (RNI) is the daily intake that meets the nutrient requirements of almost all (97.5%) individuals of the general population (in the respective age- and gender-specific group); ^b^ g/kg body weight per day; ^c^ Adequate Intakes (AI); ^d^ kilojoule/day according to moderate physical activity; * Upper 65 year: calcium = 1200–1300 mg/day, vitamin D = 10 μg/day, iron = 8 mg/day, pyridoxine = 1.7 (female), 1.5 (male) mg/day recommended; ^e^ energy requirements are based on the average energy required for people of a healthy weight who are moderately active; ^g^ mg equivalent/day of retinol; ^s^ vitamin D in the absence of endogenous synthesis µg/day; ^p^ premenopausal women/postmenopausal women; ^t^ percentage of total energy intake; ^l^ above 30 years of age; ^Δ^ = population reference intake (PRI), if unavailable, acceptable intake (AI) was used; ND = not determined, M = men, W = women.

## Data Availability

Not applicable.

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
