# Peer review of "Dietary Intake of Adult Residents in Luxembourg Taking Part in Two Cross-Sectional Studies—ORISCAV-LUX (2007–2008) and ORISCAV-LUX 2 (2016–2017)"

_nutrients, 2021, doi:10.3390/nu13124382_

Round 1

Reviewer 1 Report

Authors took into account all my comments (in the first review). The manuscript is easier to read and understandable.

I have not more comments.

best regards

Author Response

Reviewer 1:

Authors took into account all my comments (in the first review). The manuscript is easier to read and understandable. I have not more comments.

Reply: We thank the esteemed reviewer for reviewing this article.

Reviewer 2 Report

This paper compared the dietary habits of men and women in Luxembourg using a dietary FFQ questionnaire from around 2007 and another one ten years later.  Results showed changes in patterns of eating to a more detrimental pattern that includes higher saturated fat, more alcohol, and decrease in plant diversity, suggesting this pattern being influential on rising rates of non communicable diseases.

There were some great changes instituted in this version--for example the flow chart of people included and excluded.  This is a paper heavy with information. 

I only have a couple of notations:

1) on page 4, authors did state the differences between the two surveys by identifying how many different questions on various food groups existed.  I'd include ONE sentence that highlights what these differences were.  For example, what did that one more carbohydrate source represent on that newer questionnaire?

2) authors need to verify that Kolmogorov is the correct test--it is my understanding the Shapiro wilks is a better normality test with populations under 2000.

3) In the Tables, the left most column will look much better if it is left justified rather than centered.

Author Response

Reviewer 2:

This paper compared the dietary habits of men and women in Luxembourg using a dietary FFQ questionnaire from around 2007 and another one ten years later.  Results showed changes in patterns of eating to a more detrimental pattern that includes higher saturated fat, more alcohol, and decrease in plant diversity, suggesting this pattern being influential on rising rates of non-communicable diseases. There were some great changes instituted in this version--for example the flow chart of people included and excluded. This is a paper heavy with information.

Reply: We thank the reviewer for these encouraging and constructive comments.

I only have a couple of notations:

1) On page 4, authors did state the differences between the two surveys by identifying how many different questions on various food groups existed.  I'd include ONE sentence that highlights what these differences were.  For example, what did that one more carbohydrate source represent on that newer questionnaire?

Reply: We have revised the section to improve clarity. In fact, more questions (options) were introduced to increase the accuracy of the FFQ in the second wave, while the remaining questions were almost the same. However, we have included this aspect as a potential limitation in the discussion, please see line 541.

2) Authors need to verify that Kolmogorov is the correct test--it is my understanding the Shapiro wilks is a better normality test with populations under 2000.

Reply:  Thank you for your comment. Actually, we have also strongly considered the graphical interpretations (Q-Q plot and box-plots), as all normality tests have their pitfalls (e.g. being prone to outliers). However; the Shapiro–Wilk test has been stated to be a more appropriate method for small sample sizes (<50 samples), although it can also handle a larger sample size while the Kolmogorov–Smirnov test is used for n ≥50 (Please see the following article:

https://www.ncbi.nlm.nih.gov/pmc/articles/PMC6350423/)

3) In the Tables, the left most column will look much better if it is left justified rather than centered.

Reply: We thank the reviewer for this comment and did the adjustment as recommended.

This manuscript is a resubmission of an earlier submission. The following is a list of the peer review reports and author responses from that submission.

Round 1

Reviewer 1 Report

The authors reported trends of dietary consumption across a decade in the form of a cross sectional series consisting of two reports.  The main contribution of this paper is a thorough examination of dietary shifts over ten years.  This is significant in light of the rising rates of chronic disease and obesity across the world. 

The paper was very well written and the statistical analyses quite thorough.  It was essentially very high quality and coherent--results and conclusions were accurate in my opinion.  My greatest pleasure in reading this manuscript was the weaving of discussion around the significance of the diet against the gut microbiome and its relationship to disease.  This is exactly correct and timely.  Every point I was thinking of to address was addressed.

I don't have too many comments for improvement--except for the following:

1) You did an excellent job addressing the gut microbiome.  I'd insert one sentence in the discussion or conclusion that it's not just about the fiber but also the diversity of plant foods that is important--this may not be met because the trend for decreased plant food consumption while increasing processed foods.  Processed foods also disrupt the gut microbiome.  Perhaps you could make a recommendation for nutritionists to educate more on plant diversity in your conclusion statements.

2) ON page 3, you present the Food Choice Process Model and Food Choice Priorities model - can you insert one sentence of explanation of what this is?

3) ON Page 10, you report Odds ratios.  I may have missed it but what are you comparing?  Maybe be a little clearer here.

4) In your Table 8, many of the US RDAs are incorrect-you may want to verify them.  For example, the Mg rda for women is 310 and 320 based on different ages, the iron is wrong or confusing.  You have 8 mg which is the recommendation for older women (this is a footnote also) But I'd recommend to insert the premenopausal women and men RDAs with a footnote of what the RDAs are for older women/men.  Fiber should be 25 mg females and 38 males. Sodium RDA is 2300, as examples.

5) Please check your references.  They are entered inconsistently.    For example some references have page numbers from the first to last page, others have only the first page and still others have the first page a dash (-) and then missing the final page.  (5,8,9 32,33,34,39 are some of the ones that popped up for me).  Also check the language.  Some of the Authors when listing as an organization have been truncated and not clear (Ref 34--I wasn't sure if this was the correct author - EaOHS).  Also check journal names--some are abbreviated, some are fully written, some have all capitals, others don't

6) Many of the tables have strange formatting (the left hand columns should be left justified not centered)

7)Chocolate is misspelled on Page 4

Reviewer 2 Report

The manuscript described the evolution of dietary intakes, nutrient intakes and dietary pattern between 2007 and 2017 among adults living in Luxembourg. The dietary assessments were based on a validated FFQ. Authors compared average food groups, nutrient intakes between the 2 waves, as well as a dietary pattern score estimated based on principal component analysis. Results showed an increase in fat and sodium (men only) and a decrease of total carbohydrates which was in accordance with an increase of animal-based foods (except dairy) and a decrease of fruits, vegetables, grains. The evolution was almost similar between men and women. A majority of nutrient intakes are in accordance with several set of nutritional recommendations.

General comment:

This manuscript is well written and interesting to understand the evolution in the dietary consumptions over 10 years. My major comment is to make the manuscript easier to read. There is a necessity to better or simpler explained the dietary pattern approach, and to homogenize the terms use to talk about food groups (fast foods was used, processed foods was used, animal-based protein, animal-based food items …). The other major comment is related to the discussion which could be more summarized especially about all the literature on health or adverse effects of food groups consumption. Also, the likely over-estimated nutrient intakes which may explained in part a high adherence to nutritional recommendations, could be discussed.

List of comments:

Abstract : the second part of the conclusion should be reformulated: “macronutrient consumption patterns” is not adapted. Also, the conclusion may focused on food groups and macronutrients.

Please add the sign “-“ when percentage characterized a decrease.

Introduction: “In Luxembourg, the dietary patterns, similar…” a reference is missing.

Introduction : I’m wondering whether “non-nutrient intake” is convenient?

Section 2.2 :

It is not clear what was the difference between FFQ for the survey conducted in 2007 and the FFQ for the survey conducted in 2017.The same reference (number 33) is used for both. The cited reference is dated to 2013. I cannot understand which FFQ was used in the first survey. The cited reference is a biomarker validation, specificity dedicated to cardiovascular disease. Your study is not focused on cardiovascular diseases, but focused on the overall nutritional quality of consumptions. It seems that another validation (same authors) is published against a food diaries.  Regarding the FFQ of the first survey (134 items), authors said “Additional miscellaneous items included jam, …” . This sentence is confusing and especially the word “additional”. What was the final number of items ? For the FFQ of the second survey (174 items), the sum of items is equal to 173 (including miscellaneous). What is the right number?  How did you link a food item to one or more foods of the CIQUAL-ANSES database? I assume that the nutritional quality of a food item is an average of nutritional composition of ciqual foods included in the corresponding food item. Is it an unweighted average?

The paragraph “ To calculate the dietary pattern ….. Western dietary pattern (Table1)” should be moved after the explanation of the PCA. Please clarify, which food groups are considered as healthy and which food groups are considered as Western diet. Did you define a food group as healthy or Western a priori or depending on the PCA results? Also, it is not clear the way to link the 134 food items or 174 food items to 14 food groups. Does one food item classified in one food group? Or one food item decomposed in 14 food groups?. Does the method based on median and PCA and then the weighted sum to have the dietary pattern a validated approach? Or an original approach? I’m not sure that the reference 35 described the exactly same approach.

Section 2.4.

14 food groups were used in the PCA. Please, specify that intakes used was expressed in grams (this is what I assume). The PCA and FA need to be explained in simpler words. Why PCA is not sufficient to find dietary pattern. I know the principle of PCA to construct dietary pattern, but the method that you described is obvious. There is a lack of results which could be add in supplemental (see my comment below).

Section 3.0. Results

The first 3 lines is a result already showed in method section. One among both should be deleted.

The sentence “In  addition  to  being  older,  the participants in ORISCAV-LUX 2 had a higher BMI” seems to be false. Please verify.

Does the reference 36 provides different information compared to reference 35?

The descriptions of the results on PCA and the translation in 2 dietary pattern is not clear. Please clarify (see my comment below for supplementary figure 1).

Table 1. please remove % of women in the title of the columns. The column “variable” should be aligned to the left. Dietary patterns scores should move to the table 4. If I well understand, this score is a sum weighted by coefficient indicated in columns Rotated Component Matrix of table 4.

Table 2. and Table 3

In the title, ”distribution” should be replaced by “Means and SD”

You may use H2O instead of total water because it is confusing with water as a beverage.

A general question: you showed that the 2007 population was significantly different from 2017 population based on socio-demographic variables (table 1). So in table 2, you should compared nutrients intakes adjusted on socio-demo confounding factors, to catch the net time effect. Why did you not adjust?

Table 4.

In the title, ”distribution” should be replaced by “Means and SD”

The column “food groups” should be aligned to the left.

A footnote was used for “fast food”. It is important to keep all along the manuscript the same terminology and the same name of food groups.

Table 5.

The gender effect could be tested using interaction gender*time in the logistic and linear model. The age effect was investigated for nutrient intakes but not for the dietary patterns.

Table 8

Means nutrients intake by gender are already showed in tables 6 and 7. They should be removed in table 8. Table mays show the percentage adequacy to the different set of recommended values. The recommended values could be displayed in a supplementary table.

Section 4. discussion

The term “simple sugar” appears for the first time in the discussion!!

In the discussion, authors discuss results in percentage but these results are not shown inside the manuscript (same comment for abstract). A column should be added in tables in order to present percentage variable between 2007 and 2017.

“Especially  the  higher  intake  of  sugar-sweetened beverages, in conjunction with the higher intake of processed and meat foods,…” processed meats is not understandable.

“Regarding  dietary  and  soluble  fiber,  since  the  intake  of  whole grains, fruits and vegetables was significantly reduced, a trend for a decreased intake was observed.” The explanation and interpretation is not relevant. How to explain the stagnation of fiber while grains and F&V decrease. The contribution of food groups to nutrients intakes may help.

“This contradictory result could be due to the intake of other beta-carotene sources  such  as  starchy  vegetables,  including  carrots,  cabbages,  and avocado,  which  in  our  study  showed  a  significant  increase” Please check this sentence. Starchy foods decreased.

“Somewhat contrarily, Macdiarmid  et  al.  in  a  study  in  the  UK  reported  that,  associations between dietary sugar and fat intake and BMI were different between men and women. They concluded that, high fat and sweet products, might partly explain BMI increase in women”. This sentence is difficult to understand.

The term “whole grain” is used in the discussion but it did not appear in the result section.

General question : why author did not use a score estimating the adequation of food consumptions to dietary guidelines as PNNS-G-Score for example?

Figure 1 could be moved in supplementary results or removed.

Supplementary figure 1. This supplementary figure should be upgraded including results of PCA on the first 4 axes : representation of the 14 food groups, and how it is converted into 2 dietary patterns (healthy and Western). Each food group is associated to only one dietary pattern. Is it a predictable results? Could a group be associated to both dietary patterns?

Reviewer 3 Report

This study compared nutrients intake pattern collected from SQ-FFQ among adults in Luxembourg (2007-2008 vs. 2016-2017). It would be important to know the changes of nutrients intakes among people during the past decades for establishing optimal healthy dietary guideline. However, several things need to be elucidated and more explained to support the aim of the study. 

[main concern]

Q1) Nutrients intake data were collected from SQ-FFQ. However, the numbers of food items examined in the 2007-2008 SQ-FFQ (134 items) are different from those examined in 2016-2017 SQ-FFQ (178 items). We assumed that the data collected in 2016-2107 might be more precisely examined than that in 2007-2008, thereby (it is possible), the caculated amounts of nutrients intake in 2016-2017 being much bigger than those in 2007-2008. How did the authors control this possible kind of bias?  Please add more explanation on this matter. 

Q2) The authors used the median intake of major food groups to calculate the dietary pattern score. However, the weight on each food group may be differently applied.  Did you apply the weight on each of food groups?  In addition, the authors explained that "scores ranged from -8.4 to +7.8, with higher scores representing a healthy dietary pattern and lower scores representing a Western dietary pattern (Table 1)". Please explain it clearly and more in details.  

Q3) To know the change of nutrients intake pattern is important because it is closely related to the incidence and prevalence of diseases such as obesity, metabolic syndrome, diabetes, cardiovascular disease etc. To clarify the aim of the study, the authors need to examine if the changes in the incidence or prevalence of nutrition related diseases are related to the changes of nutrient intake pattern during the decades.   

[minors]

English editing is needed throught the manuscript (in grammar and sentences)